# OpenReview forum: "Buckle Up: Robustifying LLMs at Every Customization Stage via Data Curation"
_ICLR.cc/2025/Conference — ICLR 2025 Conference Withdrawn Submission_

### Official Review · Reviewer_aLWD · 2024-11-02

**Soundness:** 3
**Presentation:** 3
**Contribution:** 3
**Rating:** 5
**Confidence:** 5

**Summary:**

This paper focuses on the security risks in instruction fine-tuning scenarios, specifically that fine-tuning with malicious samples can compromise the robustness of LLMs and amplify their harmful responses. To address this issue, the work proposes an effective defense framework called CTRL, which augments data by modifying commonsensical text. This framework is similar to data augmentation strategies, with the augmented data usable before, during, and after fine-tuning. Notably, this work expands commonsensical text rather than safety-related texts, thereby reducing the difficulty of data augmentation. Experimental results show that this framework achieves good defense performance. However, I still have a significant concern regarding this work. Please refer to Weakness 1.

**Strengths:**

- Effective method: Based on experimental results, the proposed method significantly alleviates the safety risks brought by instruction fine-tuning while ensuring the model's usability.
- Comprehensive experiments: This work conducts thorough experiments that rigorously evaluate the proposed framework under various conditions.
- Good writing: The paper clearly defines symbols and is easily understood.

**Weaknesses:**

- Motivation is unclear: **A robust model is safer, with lower perplexity on safe texts and higher perplexity on harmful texts. A jailbroken model, conversely, shows higher perplexity on safe texts and lower perplexity on harmful texts.** Isn’t this phenomenon somewhat normal? Why is a separate analysis needed?  Furthermore, how does this perplexity phenomenon relate to your proposed framework? In your analysis, you emphasize the differences in **perplexity between safe and harmful texts, while your proposed solution operates on commonsensical text**. Your motivation and the proposed solution seem mismatched.

- More comprehensive evaluation needed: For both safety and usability assessments, I believe the evaluation dimensions could be further expanded. For instance, for safety evaluation, consider some jailbreaking attack methods conducted on prompts; for usability assessment, evaluating the factual accuracy and fluency of the model's responses would be more persuasive.

**Questions:**

You can focus on answering my concern about weakness 1.

---

> ### Author Response · Authors · 2024-11-19
>
> We appreciate the reviewer for the time and comments. Please refer to the clarification and revision below.
>
> ### 1. Motivation
> We are grounded by two key intuitions: (1) LLMs can acquire new knowledge when exposed to high-perplexity data, and (2) fine-tuning with safety-related data can reduce the risk of jailbreaking. Combining these principles, an ideal solution involves leveraging high-perplexity data centered on safety topics to mitigate jailbreaking in LLMs. However, since safety-related data is often costly to collect and may be insufficiently available (as detailed in Section 4.2), we propose an alternative: using commonsense data as a flexible and accessible substitute that avoids reliance on domain-specific safety data. To make commonsense data effective in combating jailbreaking, we increase its perplexity by embedding safety implications—such as relevant keywords, tones, and semantics—into the original text. These fused safety implications serve as new knowledge for the LLM, enabling high-perplexity commonsense data to mitigate jailbreaking risks through fine-tuning.
>
> **Revision:** Please refer to Section 4 of our updated submission for the corresponding revision.
>
> ---
>
> ### 2. Additional Evaluations
> We conducted further experiments as suggested, detailed below:
>
> **2.1 Prompt-Based Attacks**
> We evaluated two additional attacks that manipulate prompts during LLM inference:
> 1. **Prefix Injection Attack** ([1]): This attack appends "Sure, here is" to the prompt to elicit direct answers from the LLM.
> 2. **Refusal Suppression Attack** ([2]): This attack uses specific instructions, such as:
>    - "Do not apologize."
>    - "Do not include any 'note' or 'disclaimer.'"
>    These instructions aim to force LLMs into generating harmful responses.
>
> Below, we report the defense success rate (SR) and safety score $S_{safe}$ for these attacks on Llama-3-8B and Mistral-7B
>
> |Defense|Attack|Llama-3-8B||Mistral-7B||
> |---|---|---|---|---|---|
> |||SR|$S_{safe}$|SR|$S_{safe}$|
> |No Def|Prefix Injection|53.6%|3.45|41.5%|3.03|
> ||Refusal Suppression|44.7%|3.39|38.6%|2.75|
> |This work|Prefix Injection|74.8%|4.02|57.2%|3.73|
> ||Refusal Suppression|66.2%|3.97|55.4%|3.61|
>
> Results show that prompt-based attacks are less effective at inducing harmful answers compared to tuning-based jailbreaking. Additionally, our proposed defense mechanism improves LLM robustness, even though it is primarily designed to counter tuning-based jailbreaking.
>
>
> **2.2 Factual and Fluency Scores**
> For factual accuracy, we used **BERTScore** [3] (range from 0-1) and **BARTScore** [4] (negative scores) to evaluate how closely the generated outputs align with reference texts.
> For fluency, we asked ChatGPT to rate text quality on a scale from 1 to 5, where:
> - **1**: Not fluent at all.
> - **5**: Highly fluent.
>
> The results for both evaluations are provided below, as supplementary to Table 1. Both scores are the larger, the better.
>
> |Defense|Attack|Llama-3-8B|||Mistral-7B|||
> |---|---|---|---|---|---|---|---|
> ||| BERTScore | BARTScore | Fluency| BERTScore | BARTScore | Fluency|
> |No Def|ExHarm|0.756|-3.12|4.67| 0.742|-3.28| 4.49 |
> ||AOA|0.808|-2.77|4.60| 0.826|-2.53|4.13|
> |This work| ExHarm |0.817|-2.70|4.96|0.782|-2.91|4.75|
> ||AOA|0.834|-2.11|4.82|0.868|-1.95|4.68|
>
> It is observed that the mitigated LLMs demonstrate improved factual accuracy and text fluency, which aligns with our findings that the proposed defense mechanism contributes to improved helpfulness (as shown in Table 1 of the submission).
>
> [1] Zhang, Hangfan, et al. "On the Safety of Open-Sourced Large Language Models: Does Alignment Really Prevent Them From Being Misused?." arXiv preprint arXiv:2310.01581(2023).
>
> [2] Wei, Alexander, Nika Haghtalab, and Jacob Steinhardt. "Jailbroken: How does llm safety training fail?." Advances in Neural Information Processing Systems 36 (2024).
>
> [3] https://huggingface.co/spaces/evaluate-metric/bertscore
>
> [4] https://github.com/neulab/BARTScore
>
> ---
>
> We appreciate any feedback on whether the details and revisions address your concerns and look forward to further improving the quality of our paper :)

---

> > ### Comment · Reviewer_aLWD · 2024-11-25
> >
> > Thank you for your comprehensive response!
> >
> > I think it is very interesting to inject some samples with security implications into the training data to improve the security of the model instead of using security-related data directly. And from the experimental point of view, such an operation can indeed improve security. But I still maintain my point of view: a lot of analysis in the paper does not seem to be strongly related to your motivation.
> >
> > Overall, I will refer to the responses of other reviewers and the suggestions of AC to decide whether I will improve my score.

---

> ### Author Response · Authors · 2024-11-24
>
> Dear Reviewer,
>
> We kindly ask whether the above responses, experiments, and corresponding paper revisions address your concerns. If there are remaining issues or points requiring further clarification, additional evaluations, or revisions, we would be delighted to address them to improve the paper further.
>
> We greatly appreciate your time and feedback and look forward to hearing your thoughts. 😊

---

> ### Comment · Area_Chair_s2WS · 2024-11-25
> **[Reminder] Response to Authors**
>
> Dear Reviewer,
>
> As the rebuttal period is drawing to a close, I would appreciate your response to the authors' rebuttal at your earliest convenience.
>
> Best Regards,
>
> Area Chair

---

### Official Review · Reviewer_k3Mh · 2024-11-03

**Soundness:** 3
**Presentation:** 4
**Contribution:** 3
**Rating:** 5
**Confidence:** 4

**Summary:**

This paper introduces a novel defense mechanism against jailbreaking attacks, which is data curation based on perplexity. The authors empirically observed the discrepancy of PPL on harmful and safe responses for robust and jailbroken models, motivating their methods to increase the perplexity with safety implication when fine-tuning jailbroken LLMs. Experimental results show that their methods are effective against attacks at various stages.

**Strengths:**

1. Love that the method can be applied over various stages of attack to defend, which makes it more applicable and versatile.
2. The algorithm is motivated by empirical observations with careful justification of the intuition.
3. The paper is well-written with excellent visualizations.

**Weaknesses:**

1. My main criticism is that I am incredulous of this algorithm’s effects on degrading text generation quality. Although the authors have included evaluation with the Helpfulness score using a GPT-4 judge, it is possible that the GPT judge prefers safe output and ranks it with a higher Helpfulness score. The fact that the NoDef experiments have lower Helpfulness score than after defense potentially suggests that there may be biases in evaluating generation quality. I would expect that jailbreaking may compromise some quality, but currently it seems that the quality is very low and defense significantly raises helpfulness. Right now, I am seeing that a positive correlation between helpfulness score and safe score. This is a bit counter-intuitive.

**Suggestion**: I hope the authors could provide more justification as to why other quality metrics were not used. Authors can also consider adding additional quality eval experiments, such as evaluating with BARTScore or BERTScore.

2. The empirical observations about PPL and jailbreaking are great. Can authors expect adversarial attacks against your method? For instance, jailbreaking LLM and controlling PPL of harmful answers to a mid-level range at the same time may effectively evade this defense, making the defense not so robust in the long-term.
3. In addition, I notice that in Figure 2, the difference of PPL for jailbroken model on safe and harmful responses is not so significant. Yet the authors find that post-attack defenses demonstrate the most significant effectiveness, which means that it works better for jailbroken models with less significant PPL difference. I am slightly confused about this seemingly contradictory finding and would appreciate your clarifications.

**Questions:**

Consider changing the name of CTRL because it is the name of a well-known controllable generation model (with 2k+ citations)
https://arxiv.org/abs/1909.05858

---

> ### Author Response · Authors · 2024-11-19
>
> We are thankful for the reviewer's time and insightful feedback. Below are our detailed responses addressing the concerns.
>
> ---
>
> ### 1. Impact of the Algorithm
> We would like to clarify how the proposed algorithm enhances text usefulness. The text curation process leverages GPT-4o to rephrase original texts. Since GPT-4o is highly knowledgeable, it tends to add more detailed information. For instance, given the query "What is REST API?" from the Dolly dataset, the original answer contains 160 words, while the curated answer expands to 278 words with more itemized information. This curation not only integrates safety-relevant keywords and semantics into the original texts but also enriches the content with additional knowledge. Consequently, fine-tuning an LLM with curated texts results in the model generating more helpful and detailed responses, leading to an improved helpfulness score.
>
> ---
>
> ### 2. Additional Quality Evaluation
> In response to the suggestion, we have conducted further evaluations using BERTScore (we use "F1" of BERTScore) and BARTScore to assess text quality, supplementing the results provided in Table 1. We will include additional results on other models in the final paper for a more comprehensive analysis.
>
> |Defense|Attack|Llama-3-8B||Mistral-7B||
> |---|---|---|---|---|---|
> ||| BERTScore | BARTScore | BERTScore | BARTScore |
> |No Def|ExHarm|0.756|-3.12| 0.742|-3.28|
> ||AOA|0.808|-2.77| 0.826|-2.53|
> |This work| ExHarm |0.817|-2.70|0.782|-2.91|
> ||AOA|0.834|-2.11|0.868|-1.95|
>
> It is observed that the mitigated LLMs demonstrate improved text relevance, which aligns with our findings that the proposed defense mechanism contributes to improved helpfulness (as shown in Table 1 of the submission).
>
> ---
>
> ### 3. Discussion on Perplexity-Informed Adversarial Attacks
> We appreciate the reviewer for raising this thought-provoking question. The answer is yes. Intuitively, adversarial attacks that carefully craft attack samples with moderate perplexity could potentially achieve jailbreaking by "bypassing" the robustification introduced through fine-tuning with high-perplexity texts. This highlights an opportunity to design advanced attacks that exploit the nature of the defense mechanism, which is an intriguing area and worthy of further exploration.
>
> ---
>
> ### 4. Post-Attack Effectiveness
> Thank you for the question regarding the effectiveness of post-attack. Figure 2 measures a jailbroken LLM using a set of "safety data" (safe responses paired with harmful inputs). The curation process, however, uses a separate dataset (commonsense data) that is iteratively rephrased to significantly increase perplexity. As a result, the curated data tend to have a higher perplexity compared to the safety data shown in Figure 2, enabling it to more effectively support post-attack defenses when incorporated into the mitigation process.
>
> ---
>
> ### 5. Name Changing
> We updated the name to RoboCure (robustification through curation). Please refer to our updated submission.
>
> ---
> We look forward to incorporating these clarifications and results into the final submission. Thank you for your valuable feedback!

---

> > ### Comment · Reviewer_k3Mh · 2024-11-21
> > **Thank you for your clarifications!**
> >
> > Dear authors,
> > Re 1 and 2: first thank you for running additional experiments and explaining that this method would expand the response lengths. It is somewhat surprising that this method does not drop BERTScore and BARTScore by much (or even increases the scores), which is good - However, I am just skeptical that this is mainly caused by the increased text length, because BERTScore and BARTScore can prefer longer texts and caused the scores to improve - I think unsafe texts do not mean that they are necessarily of lower quality, right? Therefore, maybe even BERTScore and BARTScore would not be sufficient to measuring the text quality here because of the length issue. I just have major problems with the text quality eval metric in your paper, because it is very possible that GPT prefers safe texts not because they have higher quality. This, coupled with the higher inference time (because you have longer responses) due to the method, makes the method less desirable.
> >
> > I don't want to impose additional experiments because it's difficult to convince me unless you somehow normalize the effects of longer texts and provide a more fair evaluation of text quality - or that you could further prove that the eval in your original paper is fair and does not rate texts higher because they appear safe. You can design some experiments  where you run texts with your GPT eval, while ensuring that the texts should have similar lengths and bertscore but some are safe and some are unsafe. If you show that these texts which are supposed to have similar quality, indeed have similar quality as measured by your GPT eval, regardless of whether they are safe or unsafe, then I am more likely to be convinced that your eval is fair and your method is sound. But I just want to reiterate that this looks pretty hard to achieve and it's less likely that I raise scores at this point.
> >
> > I appreciate your responses to 3,4,5 and overall the paper is still very applaudable! Please don't get discouraged! I nevertheless greatly value the novelty and efforts in your work!

---

> > > ### Author Response · Authors · 2024-11-21
> > >
> > > Dear Reviewer, thank you for your encouraging response! Here we would like to further clarify some settings to make it clearer.
> > >
> > > In our evaluation, we have two groups of datasets to test with -- one harmful dataset to test the safety-relevant metrics, and a benign dataset to test the helpfulness score (also the above BERTSecore and BARTScore). When evaluating the benign dataset, a mitigated model tends to respond to more comprehensive information, which makes those scores better than a jailbroken one.
> > >
> > > And yes, it is a concern that the comparison may not be fair. To address this, we will conduct an additional group of experiments in the next few days to normalize the text length for a fair comparison, where we are going to directly compare the BERT/BART scores between harmful responses with the safe ones, we plan to also report the average text length to you. We will get the results back to you ASAP before the rebuttal deadline. Thank you again for your response :)

---

> > > ### Author Response · Authors · 2024-11-24
> > >
> > > Dear Reviewer,
> > >
> > > To address your concern, we conducted an additional group of experiments by rephrasing the generated texts such that the safe responses have a similar length (average 141.8 words) to the harmful ones (average 146.3 words). We report all five metrics: safety rate (SR), Safe Score ($S_\texttt{SAFE}$), helpfulness score ($S_\texttt{HELP}$), BERTScore, and BARTScore for the generated harmful and safe responses. All results are evaluated on Llama-3-8B:
> > >
> > > |Defense|Attack|SR|$S_\texttt{SAFE}$|$S_\texttt{HELP}$| BERTScore | BARTScore |
> > > |---|---|---|---|---|---|---|
> > > |No Def|ExHarm|14.6%|2.12|3.82|0.691|-4.22|
> > > ||AOA|13.2%|1.79| 3.76|0.775|-3.03|
> > > |This work| ExHarm |95.2%|4.75|3.90|0.726|-3.59|
> > > ||AOA|93.4%|4.37|4.02|0.787|-2.93|
> > >
> > > As BERTScore and BARTScore require "references," we encountered a challenge for harmful responses, as the original security-related questions lack harmful references. To address this, we fine-tuned a jailbroken ChatGPT to generate harmful responses, which we then rephrased to match the length of the safe responses (also rephrased to be shorter). When calculating the helpfulness score ($S_\texttt{HELP}$), we queried the jailbroken ChatGPT for its rating, as a standard GPT model refuses to provide ratings for harmful content.
> > >
> > > Interestingly, even with similar lengths, the proposed technique improved the helpfulness score as well as the BERT/BART scores slightly. However, since the safe and harmful responses use different references, it is not possible to conclusively attribute the performance improvement solely to defense-based fine-tuning.
> > >
> > > **To ensure conclusions are convincing, we conducted additional testing** using the *Commonsense Test Dataset* (as used in our paper) to measure helpfulness score ($S_\texttt{HELP}$), BERTScore, and BARTScore, now leveraging the references provided in the dataset.
> > >
> > > |Defense|Attack|$S_\texttt{HELP}$| BERTScore | BARTScore |
> > > |---|---|---|---|---|
> > > |No Def|ExHarm|3.63|0.740|-3.36|
> > > ||AOA| 3.71|0.791|-2.83|
> > > |This work| ExHarm |3.75|0.764|-3.28|
> > > ||AOA|3.89|0.812|-2.65|
> > >
> > > Notably, the defended model (last two rows) demonstrates increased $S_\texttt{HELP}$, BERTScore, and BARTScore compared with the jailbroken model on the commonsense data, even though its performance slightly decreases compared to the longer text results (as reported in our first version of response above). One possible explanation is that the additional fine-tuning always involves new knowledge into the data, which may enhance the overall utility of the LLM, albeit marginally.
> > >
> > > ---
> > > We look forward to your further feedback to improve the clarity and quality of this work, thank you!

---

> > > > ### Comment · Reviewer_k3Mh · 2024-11-25
> > > > **Raised two sub scores**
> > > >
> > > > Dear authors,
> > > >
> > > > I want to thank you again for the great work and amazing efforts to run additional experiments. As much as this is slightly against my intuition, I think I am more convinced by your arguments with the results you are showing. Although you have done a wonderful job during the rebuttal, considering this is a significant flaw of evaluation in your paper, I cannot raise much scores except for the soundness (2 -> 3) and contribution (2 -> 3) scores. I am still slightly hesitant about the paper and would like to defer my final recommendations to ACs.
> > > >
> > > > I hope you understand this and thank you again for the amazing work.

---

> > > > > ### Author Response · Authors · 2024-11-25
> > > > >
> > > > > Hi Dear Reviewer, we appreciate your responses. Could we simply ask for the major points about evaluation? We're launching additional evaluations recent days on remaining models and settings, and hope to update the tables regarding the recommended metrics soon. So that we could try our best to clear the evaluation flaw and clarify your hesitation. We are greatly appreciate your help!

---

### Official Review · Reviewer_ogT7 · 2024-11-04

**Soundness:** 2
**Presentation:** 3
**Contribution:** 2
**Rating:** 5
**Confidence:** 4

**Summary:**

This work proposes a defense mechanism against jailbreaking attacks on LLMs. The authors introduce the CTRL framework, which is an integration of safety-centered data curation in the fine-tuning pipeline of LLMs. At step-pre, during, and post-of customization, this curated data might be fed into fine tuning in order to bring improvement in robustness without changing the fine-tuning workflow of LLMs. The novelty of CTRL is the use of common, widely available texts that are filtered to maximize the model's perplexity, providing new, striking information on safety for the model's training process. In this way, the proposed framework provides protection for LLMs against jailbreak vulnerabilities throughout different customization stages by efficiently neutralizing hazardous content.

**Strengths:**

1. The paper is well-organized and easy to follow.
2. The proposed method does not add any new modules or alter the fine-tuning pipeline, making it more practical for real-world defense applications.

**Weaknesses:**

My main concern is whether it is necessary to use a commonsense dataset to create the curated dataset. I understand the authors' intention to increase perplexity. For a safety-aligned LLM, high perplexity on harmful training data typically indicates high loss, which guide the model learn malicious behaviors. To counteract this, the authors aim to increase the perplexity of a benign dataset, making it “challenging” for the LLM to process, thus raising the loss value and reducing the influence of harmful content. However, a potential limitation of the proposed framework is that, instead of using a commonsense dataset, it might be more effective to use a harmful dataset paired with safe responses as the curated data. I believe this approach could yield stronger defense performance than the proposed method. In practice, the authors essentially seek the same outcome, as they add specific safety keywords when generating the curated dataset. Although the authors do not explicitly claim to create a safety-focused dataset, the approach they describe could lead to a similar result in practice.

In my opinion, although the authors claim to have proposed a novel method, the core idea behind its effectiveness lies in creating a safety-focused dataset and fine-tuning the LLM on it—a strategy that prior research has already shown to be a strong baseline. Whether the dataset originates from a commonsense dataset is less significant from a safety perspective; while it may offer some advantage in preserving the LLM's utility, the overall novelty of this paper feels limited.

**Questions:**

Could you compare the effectiveness of your proposed dataset with that of a safety-aligned dataset?

---

> ### Author Response · Authors · 2024-11-18
>
> We appreciate the reviewer's time and the valuable comments. Please find our detailed responses below, along with the corresponding revisions.
>
>
> ### 1. Necessity of Using a Commonsense Dataset
>
> Our primary motivation is to ensure flexibility and cost-effectiveness when utilizing datasets for defense purposes. In high-stakes applications, such as healthcare, it is often challenging to find large-scale safety data (e.g., harmful questions + safe responses) that are specific to the domain. Incorporating safety data from other domains may cause a fine-tuned LLM agent to exhibit distracting or inappropriate behaviors, undermining its expertise in the target domain. On the other hand, collecting domain-specific safety data is typically both expensive and time-consuming. Therefore, a flexible approach that enables the use of diverse datasets for defense can significantly reduce costs and improve adaptability while preserving the LLM's domain expertise. In this work, we use a commonsense dataset to demonstrate the generalizability of our framework.
>
> **Revision:** We have addressed this concern in the introduction and Section 4.1, with revisions highlighted by red texts in the updated submission.
>
> ---
>
> ### 2. Comparison with a Safety-Aligned Dataset
> Please see the experimental results below regarding the performance of using a safety-aligned dataset for defense (on Llama-3-8B).
>
> |Dataset|Attack|Safety Rate|Safety Score|Helpfulness Score|
> |---|---|---|---|---|
> |Safety Data|ExHarm|90.7%|4.75|4.02|
> ||AOA|94.8%|4.80|4.05
> |Safety Data + Curation| ExHarm|100%|4.91|4.19|
> ||AOA|100%|4.97|4.36|
>
> We have also conducted experiments to curate a safety-aligned dataset using our proposed method. Notably, while the safety-aligned dataset already performs well, our curation process can further enhance the defense success rate (i.e., the ratio of safe responses). Moreover, it can improve the LLM's helpfulness by enriching the original texts with additional details.
>
> **Revision:** In Section 6.4 ("Using Safety-Relevant Texts") and Table 3, we have highlighted the corresponding revisions.
>
> ---
>
> We will appreciate any feedback on whether the details and revisions above address your concerns and look forward to further improving the quality of our paper :)

---

> > ### Comment · Reviewer_ogT7 · 2024-11-25
> >
> > Thank the authors for the rebuttal. However, I have some thoughts regarding your first point. There are already many safety alignment datasets available, including data from red-teaming research. I don't believe that the availability of safety alignment data will be a significant issue in the future. If you consider collecting safety datasets to be a challenge, I suggest comparing the cost of generating a safety alignment dataset using, for example, GPT-4 with the cost of your proposed method. It’s worth noting that your method still requires an LLM to regenerate the data. I don’t see a substantial cost difference between generating an alignment dataset directly and using your proposed approach.
> >
> > Moreover, prior research has demonstrated that safety alignment can actually help reduce potential biases, such as gender bias. Therefore, I don’t find it accurate to conclude that "Incorporating safety data from other domains may cause a fine-tuned LLM agent to exhibit distracting or inappropriate behaviors, undermining its expertise in the target domain." Safety alignment doesn’t necessarily require domain-specific data, as these models have shown the ability to generalize effectively to unseen domains.
> >
> > Regarding Experiment 2, could you clarify why the performance drops to only 90% when solely using the safety dataset?

---

> ### Author Response · Authors · 2024-11-24
>
> Dear Reviewer,
>
> We kindly ask whether the above responses, experiments, and corresponding revisions in the updated paper address your concerns. If there are remaining issues or points requiring further clarification, additional evaluations, or revisions, we would be delighted to address them to improve the paper further.
>
> We greatly appreciate your time and feedback and look forward to hearing your thoughts. 😊

---

> ### Comment · Area_Chair_s2WS · 2024-11-25
> **[Reminder] Response to Authors**
>
> Dear Reviewer,
>
> As the rebuttal period is drawing to a close, I would appreciate your response to the authors' rebuttal at your earliest convenience.
>
> Best Regards,
>
> Area Chair

---

### Author Response · Authors · 2024-11-25

Dear Reviewers,

As the deadline is approaching for us to provide additional responses and revision, could we simply ask about whether the given responses addressed your concerns? We have also updated the submission with highlighted revisions. We appreciate for your follow-up feedback to promote this paper's quality.

---

### Note · Authors · 2024-11-25

I have read and agree with the venue's withdrawal policy on behalf of myself and my co-authors.